# A Novel Scheduling Algorithm for Improved Performance of Multi-Objective Safety-Critical Wireless Sensor Networks Using Long Short-Term Memory

Issam Al-Nader *, Aboubaker Lasebae, Rand Raheem and Ali Khoshkholghi

Department of Computer Science, School of Science and Technology, Middlesex University, The Burroughs, London NW4 4BT, UK; a.lasebae@mdx.ac.uk (A.L.); r.h.raheem@mdx.ac.uk (R.R.); a.khoshkholghi@mdx.ac.uk (A.K.)
* Correspondence: ia287@live.mdx.ac.uk

**Abstract:** The multiple objective optimisation (MOO) challenges encountered in the context of wireless sensor networks (WSNs) present a formidable NP-hard problem. These issues primarily arise from the constraints imposed by critical factors such as connectivity, coverage, and, most notably, energy consumption. Simultaneously fulfilling these three requirements is no longer considered the standard approach for enhancing system dependability. To illustrate, a prospective solution may optimise one or two of these requirements while bolstering overall network energy efficiency. Nonetheless, prior endeavours documented in the extant literature reveal unexplored avenues for enhancement. Hence, this paper introduces a new methodology aimed at alleviating MOO concerns and thereby enhancing the quality of service (QoS) in WSNs. A long short-term memory (LSTM) model is proposed as an analytical tool to deliver an energy-efficient scheduling solution that aligns and optimises WSN parameters, striving to attain the most favourable system performance. The LSTM algorithm's effectiveness is assessed through the iterative application of periods, confirming the desired QoS levels. The unique feature of LSTM lies in its capability to observe specific event sequences and subsequently establish them as the system's default configuration for its entire operational lifespan. Once these favourable parameters are identified, LSTM automatically ensures consistent service availability and reliability throughout the network's lifespan. The results obtained demonstrate the superiority of the proposed LSTM-based scheduling algorithm in comparison to the self-organising map (SOFM)-based node scheduling algorithm. The LSTM-based approach outperforms the SOFM-based alternative by a remarkable 75% in terms of coverage and exhibits a 20% enhancement in network lifetime, all while maintaining equivalent levels of connectivity (i.e., 99%) in both algorithms.

**Keywords:** WSN; dependable WSN; recurrent neural network; real-time systems; QoS in WSN; SOFM





## 1. Introduction

The MOO problem has consistently been a central research focus, particularly in the realm of safety-critical WSN systems. Safety-critical systems, such as fire detection, military surveillance, and nuclear plant monitoring systems [1], are instrumental in ensuring human safety, safeguarding assets, and averting fatalities. To address the MOO problem, a novel scheduling algorithm based on LSTM was introduced to guarantee service availability and reliability. Previous analyses [2–4] revealed patterns in our coverage data, with the LSTM model employed to analyse and provide an energy-efficient scheduling solution that tackles the MOO problem. LSTM is a recurrent neural network (RNN)-based algorithm which excels at remembering short-term patterns for predicting future events. Its primary purpose in WSNs is to optimise key parameters, including energy consumption, connectivity, and coverage, to yield the best system output.

The choice of LSTM in WSNs as a proposed solution is driven by its key features [5] whereby LSTM is adept at capturing temporal dependencies and patterns in sequential data, making it ideal for sequence modelling. In WSNs, where sensor readings accumulate over time, LSTM aids in modelling and predicting patterns or events. It automatically extracts relevant features from raw sensor data, reducing the need for manual feature engineering (feature extraction). LSTM excels at capturing complex relationships and non-linearities in data, which proves valuable for anomaly detection, event prediction, and other tasks in WSNs (non-linearity). It can also handle variable-length sequences, a critical capability when dealing with irregularly sampled sensor data (flexibility).

Despite the numerous advantages offered by LSTM, several challenges should be considered when applying the model in WSNs [6]. Such challenges include the fact that LSTM can be computationally intensive, which may pose challenges in resource-constrained sensor nodes with limited processing power, memory, and energy. Collecting and storing data for training in resource-constrained WSN environments can be challenging, given the limitations of data transmission powered by sensor node batteries. Overfitting is a concern when training data are limited. Proper regularisation techniques and careful model selection are vital to mitigate this issue. Dealing with noisy or missing data from real-life sensor networks, often plagued by lossy wireless links, can be a struggle. Preprocessing and data cleaning are essential steps to address data quality problems before feeding the data into the LSTM model.

LSTM is often seen as a "black box" model, which makes it challenging to interpret the rationale behind its predictions, a factor that may be crucial in certain applications (interpretability). In the context of WSNs, assuming a sensor network can run a maximum of 2000 rounds, after each round, the LSTM model predicts the optimal configuration for the current round based on the previous round's data. This configuration represents the best sequence of nodes predicted from the previous round. This predictive process continues until a specific energy threshold is reached.

*Overview of the LSTM*

LSTM relies on the principles of training recurrent neural networks (RNNs) to mitigate the issues of vanishing or exploding error gradients. In the context of error management, if we encounter two error gradient values, say 0.1 each, their multiplication results in 0.01, signifying an increase in error. Such a scenario can lead to inaccurate predictions due to error gradient underflow or overflow. To address this, a specialised memory cell is introduced to handle short-term memory. The goal is to ensure that predictions contain minimal errors, as lower errors contribute to more accurate predictions, thereby enabling the network to improve its predictive capabilities.

In contrast to prediction, replication involves the perfect memorisation of an object with zero errors. When the error gradient approaches zero, replication becomes undesirable in this context. The primary objective in this context is to enhance the accuracy of our predictions while minimising errors. Additionally, we have a secondary goal of expanding the predictive scope by introducing a FORGET GATE parameter. To illustrate these objectives, consider a text prediction task where the aim is to generate a complete sentence like "*I want to go to London*". When we examine the error gradient, we might find that our prediction has a gradient error of 0.1, resulting in an output sentence like "*I am going to Dodnon*", where "*Dodnon*" is incorrect and should be "*London*". The focus here is to reduce the gradient error, striving for a near-optimal solution. LSTM can retain the sequence of events over extended periods, surpassing the limitations of this output reproduction. Short sequences are stored in short-term memory, while longer sequences, composed of shorter segments, are stored in long-term memory to establish valuable patterns for predictive purposes.

Continuing with the same example related to the FORGET GATE parameter, this gate becomes active once a particular pattern is finished, such as in the case of the text example, "*I want to go to London*". It then paves the way for a new pattern. If we aim to enhance our

text-prediction task by generating a longer sentence like "*I want to go to London and catch a connecting flight to Dublin to meet a friend*", we encounter two distinct parts within the sentence. By introducing the FORGET GATE, the network gains the ability to determine what to discard and what to retain. In this specific scenario, it would forget the initial part, "*I want to go to London*" while retaining the latter portion, "*and catch a connecting flight to Dublin to meet a friend*". Consequently, our prediction task becomes more efficient.

The initial sentence's processing is completed, and the LSTM is then employed to predict the subsequent sequence of the sentence, which also requires processing and prediction. LSTM classification is widely applicable in various real-life scenarios, such as text prediction, voice recognition, stock market forecasts, and the computation of WSNs.

After conducting critical analyses of previously implemented algorithms, including the hidden Markov model (HMM) node scheduling algorithm and the self-organising feature map (SOFM) node scheduling algorithm, notable observations revealed the detection of similar patterns within our coverage data. In response, we introduced an LSTM-based node algorithm to analyse and provide an energy-efficient scheduling solution that addresses the MOO problem. Scheduling solutions in this area are scarce, and the state-of-the-art solutions for addressing multi-objective optimisation have their limitations. Our originality lies in overcoming the current limitations in the existing literature, particularly in comparison to solutions such as the randomised coverage based scheduling (RCS) algorithm [3], a clique base node scheduling algorithm [4], and the HMM algorithm [2].

While there have been a few attempts to apply LSTM in the context of WSN [2] and [3] to address the primary MOO problem, these attempts also face limitations. Consequently, the main contribution of this paper is to propose a new scheduling algorithm aimed at enhancing the performance of the MOO problem in safety-critical WSNs, leveraging LSTM to address the limitations.

Therefore, this paper is organised as follows: in Section 2, we delve into the current state-of-the-art scheduling algorithms and their constraints. Section 3 outlines the problem formulation within the context of a WSN. Section 4 covers the simulation experiments and the resulting outcomes. In Section 5, the conclusion of the work is drawn.

## 2. Related Work

The literature encompasses various scheduling algorithms aimed at exploring the state space of the MOO problem, with the goal of identifying an optimal or nearly optimal solution [2,3,7–10]. These endeavours hold the promise of meeting the QoS requirements within WSNs, consequently enhancing the reliability of safety-critical systems. What follows reviews techniques that are based on temporal and spatial modes as we find them suitable for our performance evaluation.

In other segments of the literature, alternative approaches address the challenges posed by WSNs, including issues like energy efficiency [10,11], through the utilisation of LSTM models for time series prediction. A notable method in the literature [10] strives to optimise energy consumption by reducing the number of communication links between the fusion centre/base station and the sensor nodes in the WSN. This optimisation relies on a dual prediction scheme (DPS) based on a least mean square (LMS) filter to forecast the quantity of communication links between the base station (BS) and the sensor nodes.

The sensor nodes are then permitted to transmit data only when the margin of error between the sensed data and the predicted data falls below a user-defined threshold [11]. A beacon signal is then deployed within the network to instruct nodes, for both senders and receivers, to utilise the LSTM-predicted value as the current time step for sending the actual data. This strategic use of beacon signals minimises energy consumption, especially when compared to transmitting extensive data packets [11]. However, it is essential to acknowledge the limitations of this approach. Firstly, it assumes stationary nodes, which may not align with the dynamic nature of real-world WSNs. Additionally, it does not account for scenarios involving node failures, a factor that can be unrealistic in practical WSN deployments.

In [10], the authors effectively reduced energy consumption by employing an RNN-LSTM-based model. Specifically, they leveraged LSTM-based components to segment the WSN into distinct layers on the sensor nodes. This approach minimises data transmission by utilising LSTM to compare actual data with the data that need to be transmitted. Consequently, it leads to a reduction in communication overhead while also balancing the workload at the sink node. It is worth noting that the primary focus of this approach is on communication reduction, with limited attention given to coverage improvement.

In contrast, a more recent study presented in [12] introduces the utilisation of bidirectional long short-term memory (BiLSTM) techniques to offer adaptive duty-cycle scheduling for WSNs. The BiLSTM captures both past and future steps to fine-tune the sleep schedules of sensor nodes. This algorithm incorporates an eligibility check based on the Jaccard similarity index (JSI) value among the sensor nodes, electing those with higher values to enter sleep mode. The sleep schedule extension is based on data patterns, such as traffic load and energy levels, obtained through BiLSTM analysis. It is important to note that this work assumes that sensor nodes are already in a sleep mode, with intervention limited to their sleep duty cycles. However, the BiLSTM algorithm does not explicitly consider network coverage and connectivity, as it assumes a random duty cycle for all nodes. Moreover, over time, the eligibility rules impact the residual energy, thereby affecting network performance.

In [13], the author proposed an SOFM topology building (SOFMTB) algorithm to optimise energy in WSNs. Unlike LSTM, the SOFM model is a spatial technique that provides a suitable network configuration for a given problem. The topology building SOFM algorithm creates the clusters based on the energy levels for efficient dissemination of data over the network. The SOFMTB divides the WSN into tiers using a k-means algorithm as cluster head and cluster members. Thereby, the topology is created for the network, providing efficient partitions in the WSN.

In the work of [14], an SOFM was used to reduce energy consumption and the bandwidth usage in a resource-constrained WSN. The solution tends to reduce the data communication during the entire network lifecycle by reducing its size. For instance, in this work, 1500 data volumes were generated and aggregated from all sensor nodes and sent to the base station. This approach abstracts these data and reduces their size by selecting the most suitable size that includes the meaningful data and sending these instead of the full 1500 volumes.

Other research, such as that in [6], has applied LSTM models to detect distributed denial-of-service (DDOS) attacks in WSNs. Notably, this work leverages LSTM models to tackle the MOO problem within WSNs from a different perspective. The objective is to identify specific sequences of events (data) that offer optimal coverage, connectivity, and lifetime, thereby establishing a useful LSTM-based node scheduling model. Consequently, the MOO problem is recast within the context of WSNs.

## 3. Problem Formulation

This paper aims to introduce a new scheduling algorithm designed to enhance the performance of MOO in the context of service availability and reliability, which are crucial for ensuring the dependability of safety-critical WSNs. The proposed algorithm effectively manages the scheduling process and behaviour across discrete-time sequences involving time-bound multiplexing.

For instance, it divides the timeline, T, among the network's sensors, S, denoted as $T_1 = S_1$, $T_2 = S_2$, and so forth. This approach is instrumental in identifying short and energy-efficient patterns through the LSTM neural network. Consider a scenario where sensors operate with an energy level of 10 Joules, and we have a range of sensors ($S_1$, $S_2$, $S_3$, $S_4$) characterised as efficient patterns. The LSTM network then predicts the most suitable next pattern based on its training from the WSN.

The LSTM's output comprises energy functions for each range, which are subsequently utilised to project into the following cycles. This projection aids in determining the most suitable cycle to ensure both dependability and efficiency throughout the WSN lifecycle.

Prior experiments have revealed that coverage data exhibit variations in performance across different sensor zones, making it challenging for classical algorithms (such as RCS, HMM, and SOFM) to capture the optimal coverage patterns. To address this, we harnessed LSTM to capture and replicate the best coverage patterns.

Analysing the coverage from a graphical standpoint reveals fluctuations that appear random. To mitigate these fluctuations, the LSTM algorithm is employed to predict and reproduce meaningful patterns wherever possible. As a result, we propose a new scheduling algorithm specifically designed to capture valuable coverage patterns and propagate them throughout the entire WSN cycle, thereby significantly enhancing the availability and reliability of network services.

The LSTM model used in this approach incorporates various parameters, including an input $x_t$ representing the current energy level, $h_{t-1}$ denoting the previously consumed energy level, and $c_{t-1}$ as a parameter dependent on the energy level at time $t-1$. The model generates an output $o_t$ for the current time step, as well as $c_t$ and $h_t$ for the energy consumption in the subsequent time step, as shown in Figure 1.

$$f_t = \sigma_g\left(W_f \times x_t + U_f \times h_{t-1} + b_f\right) \tag{1}$$

where $f_t$ is the forget gate;

$$i_t = \sigma_g(W_i \times x_t + U_i \times h_{t-1} + b_i) \tag{2}$$

where $i_t$ is the input gate;

$$o_t = \sigma_g(W_o \times x_t + U_o \times h_{t-1} + b_o) \tag{3}$$

where $o_t$ is the output gate;

$$c_t' = \sigma_c(W_c \times x_t + U_c \times h_{t-1} + b_c) \tag{4}$$

$$c_t = f_t \cdot c_{t-1} + i_t \cdot c_t' \tag{5}$$

where $c_t$ is the cell gate;

$$h_t = o_t \cdot \sigma_c(c_t) \tag{6}$$

where $h_t$ is the hidden state.

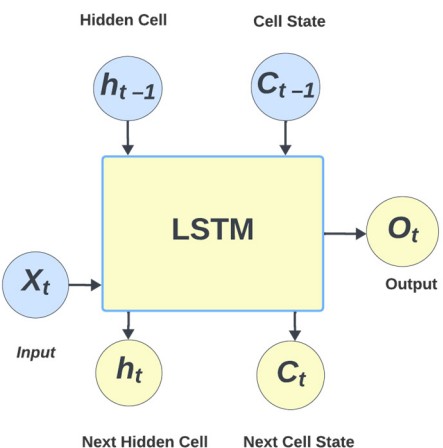

**Figure 1.** Utilisation of the LSTM model in WSN scheduling approach (mathematical representation).

LSTM leverages the current state of the network, such as the energy level of each sensor node at time t, to make real-time energy predictions for the network. This makes the solution time-dependent. LSTM, as a memory model, relies on temporal information to

predict future steps based on the current step. In contrast, SOFM is a spatial model that relies solely on the spatial configuration of the network, providing a feature-based solution.

### 3.1. The Proposed LSTM Node Scheduling Algorithm

The LSTM model operates independently and performs the training during the WSN rung time. As the WSN starts its operation, a dataset is generated, and the LSTM scheduling algorithm then starts its training process. The dataset is taken as input to the LSTM machine learning algorithm. The output dataset consists of maximum connectivity and coverage values that correspond to subsequent energy values, known as data points.

For a better understanding of the mechanism of the LSTM algorithm, suppose we have the first 10 rounds generated from the WSN with the fourth, fifth, and sixth rounds showing interesting patterns of maximum coverage and connectivity, energy, and scheduling coordinates points. Subsequently, for the next 10 rounds, these patterns of data points are identified by using the LSTM learning method and released as the output of the LSTM algorithm. The LSTM algorithm will cease to run until the threshold energy level of the sensor node falls below the operating energy level of the node.

Finally, this output is stored as a logbook and is used successfully by the same WSN without using the LSTM algorithm. Figure 2 explains the mechanism of the LSTM node scheduling algorithm.

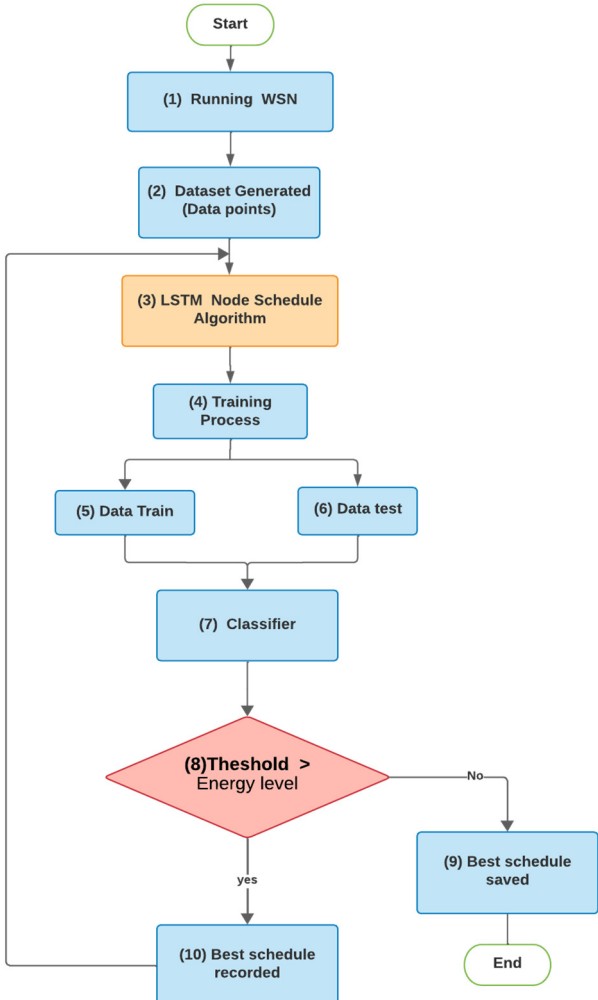

**Figure 2.** The mechanism of the LSTM node scheduling algorithm.

The process of the LSTM algorithm begins by running the network with a scheduling algorithm to collect the dataset required for the training process. Once an interesting

sequence is identified, the algorithm detects and records it. Subsequently, this sequence of interest is replicated throughout the lifetime of the WSN. Thus, after illustrating the LSTM node scheduling algorithm mechanism, Algorithm 1 presents step by step the pseudo code of the LSTM mechanism.

---

**Algorithm 1.** The process of LSTM model implementation in a WSN

---

- **Input:** Energy level values at $t = t - 1$ and $t = t$, dependable parameters (coverage and connectivity)
- **Output:** Scheduling (energy levels at $t = t + 1$ (prediction), dependable parameters (coverage and connectivity))

1: Initialise LSTM model
2: Define input shape based on sensor data dimensions
3: Define LSTM layers
4:     Add LSTM layer with specified number of units and input shape
5:     Add additional LSTM layers if necessary
6:         Add dense layers
7:         Add dense layers with appropriate activation functions
8:         Compile the model
9: Specify loss function, optimiser, and metrics
10: Train the model
11: Provide training data (sensor data and corresponding energy consumption)
12: Specify the number of epochs and batch size
13: Evaluate the model
14: Provide test data to evaluate the model's performance
15:    Use the trained model for energy scheduling
16: Collect real-time sensor data
17: Preprocess the data to match the input shape of the LSTM model
18:    Use the model to predict the energy consumption for the given sensor data
19:    Implement energy scheduling algorithm
20: Determine the energy requirements and constraints of the wireless sensor network
21:    Use the predicted energy consumption values to schedule energy usage and optimise resource allocation
22:    Consider factors such as battery life, energy efficiency, network coverage, and data transmission requirements
23: Repeat the energy scheduling process periodically
24: Collect updated sensor data
25: Preprocess the data and feed it to the LSTM model for energy consumption prediction
26: Adjust the energy scheduling algorithm based on real-time sensor data and network conditions

---

### 3.2. Implementing LSTM in WSN

LSTM stems from RNN methodology with the objective of finding meaningful patterns with respect to the design's interest and detecting such patterns using training algorithms as illustrated in Figure 3 below.

The LSTM architecture comprises several key components:

Sequential RNN Unit: The LSTM begins with an assembly of sequential recurrent neural network (RNN) units.

Cell state (LSTM): The core of the LSTM is the cell state. It serves as a memory unit that stores information from past time steps and controls the flow of information within the network.

Hidden state: A cell that is responsible for the encoding of the most recent time steps of the data.

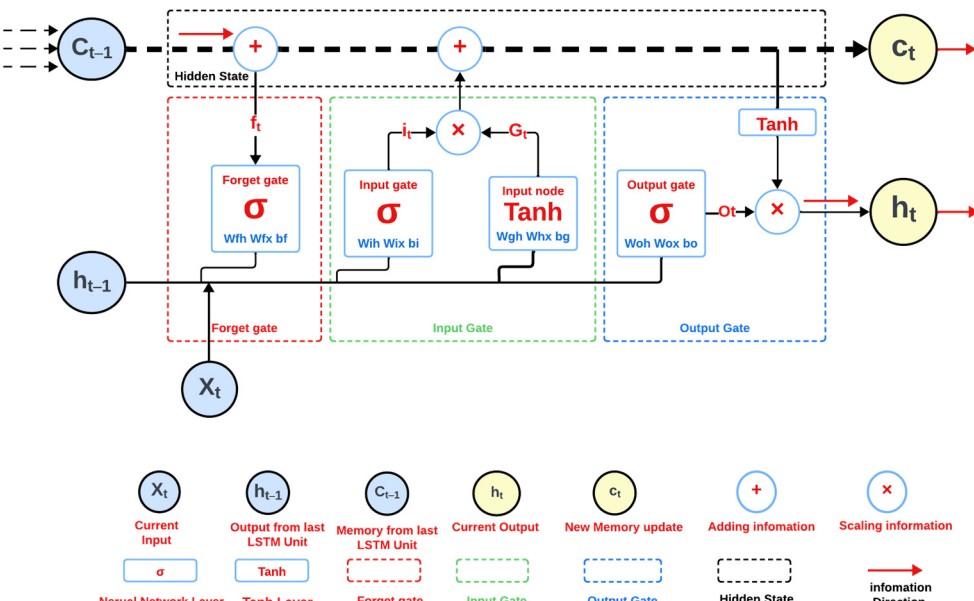

**Figure 3.** LSTM implementation in WSN.

Forget gate: This gate determines what information from the cell state should be discarded and what should be retained. It uses a sigmoid function to make a decision. Two inputs $x(t)$, i.e., an input of schedule sequence at the particular current time from the WSN and $h(t-1)$, i.e., the previous cell output of schedule sequence from the WSN, are fed to the forget gate and multiplied with weight matrices followed by the addition of bias. The result is passed through an activation function which provides a binary output. If, for a particular cell state, the output is 0, then the piece of information is forgotten, and for output 1, the information is retained for future use for the input gate.

Input gate: This gate is responsible for deciding what new information should be added to the cell state. It uses another sigmoid function for this purpose. First, the information is regulated using the sigmoid function and the values to be remembered are filtered similar to the forget gate using inputs $h(t-1)$, i.e., an input of schedule sequence at the particular current time from the WSN and $x(t)$, i.e., the previous cell output of schedule sequence from the WSN. Then, a vector is created using the tanh function that gives an output from $-1$ to $+1$, which contains all the possible values from $h(t-1)$, i.e., an input of schedule sequence at the particular current time from the WSN and $x(t)$, i.e., the previous cell output of schedule sequence from the WSN. Finally, the values of the vector and the regulated values are multiplied to obtain useful information.

Output gate: This gate controls what information from the cell state should be used to produce the output. It employs a hyperbolic tangent (tanh) function to create a vector, which is then used for the prediction. First, a vector is generated by applying the tanh function on the cell. Then, the information is regulated using the sigmoid function and filtered by the values to be remembered using inputs $h(t-1)$, i.e., an input of schedule sequence at the particular current time from the WSN and $x(t)$, i.e., the previous cell output of schedule sequence from the WSN. Finally, the values of the vector and the regulated values are multiplied to be sent as an output and input to the next cell.

The training process of the LSTM involves three sigmoid functions and two hyperbolic tangent functions. These functions are well-suited for handling data that evolve over time [11]. The training of the LSTM network results in an equation with two variables, one of which is time. This enables the network to predict one variable when the other is provided. The accuracy of these predictions serves as a measure of the LSTM network's efficiency.

In WSN applications, each round involves predicting the network configuration with a focus on optimising energy usage and minimising the distance from the sink. These

network configurations are linked to the locations of nodes and play a crucial role in achieving convergence and extending the network's lifespan.

## 4. Results and Discussion

A MATLAB simulator version R2022a was employed to conduct simulations and experiments on the network. Specifically, we exclusively compared the LSTM scheduling algorithm with the SOFM algorithm. This selection was based on SOFM's demonstrated superiority over other algorithms that were previously investigated, namely HMM [2], and RCS [3]. The SOFM node scheduling algorithm uses features to provide the schedule solution. There are five features considered in the process of the SOFM, namely: (1) the position of cluster heads with respect to the sink, (2) the distance between the parent and child nodes, (3) the energy levels, (4) the transmission link (Tx), and (5) the receive link (Rx). In the SOFM process, the Euclidean distance is calculated, and the best matching is identified where the clustering starts to form a group as a potential solution. Each group has its own scheduling which is based on a value obtained from the features. The SOFM algorithm provides a spatial customised cluster-based configuration that is associated with a certain weight value which is updated at every iteration. All these features' processes are calculated by the BS during the design time and implemented in the run time. Thus, we find it suitable to benchmark the new proposed algorithm against the SOFM for performance evaluations. The experiments were conducted using the following parameters, and all resulting data were collected over a 30-run period.

Figure 4 demonstrates the simulation experiment for the LSTM with 100 homogenous nodes, i.e., having the same capabilities (sensing, communicating, power battery, processing power). The following assumptions were considered:

1.  The WSN-monitored area in simulations is 100 m$^2$.
2.  All nodes are connected to one sink node.
3.  Nodes are randomly deployed.
4.  The number of iterations is fixed at 200 epochs.
5.  The number of simulation iterations is limited by convergence of the LSTM network when the wait time reaches a stable condition.

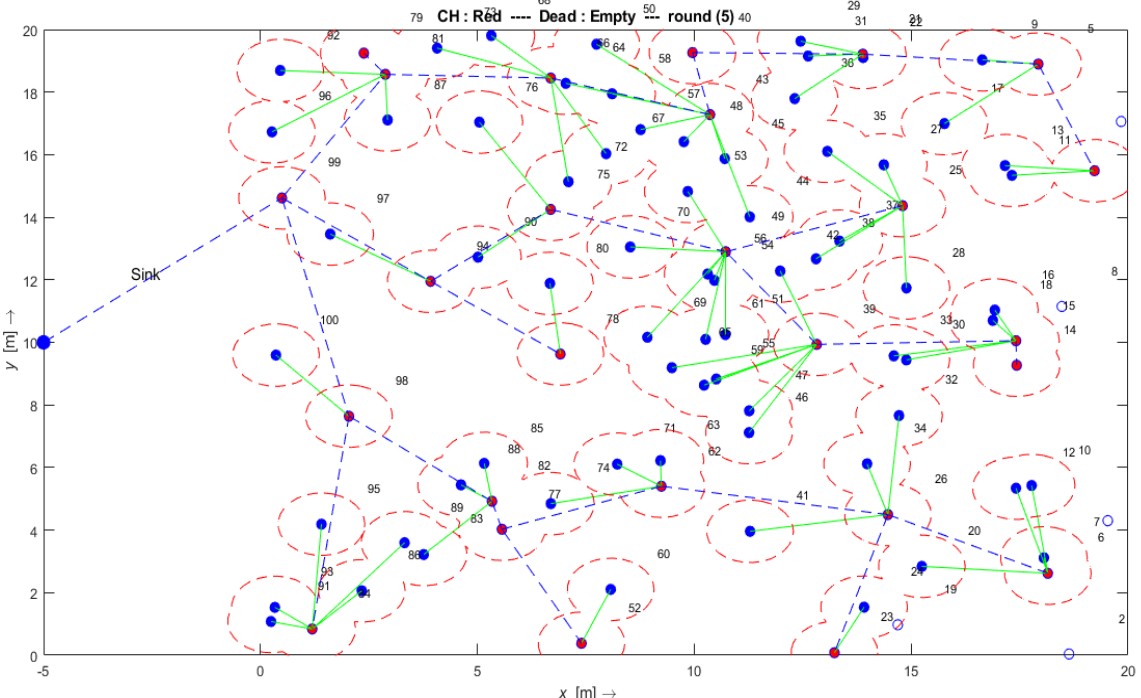

**Figure 4.** LSTM experiment in MATLAB (Red—cluster head, Blue—child node).

The simulation experiment was individually tested 30 times as mentioned above using one-way ANOVA analysis, which was implemented on the SOFM and the LSTM algorithms to identify the *p*-value for statistical results. Table 1 presents the statistical analysis of both LSTM and SOFM node scheduling algorithms.

**Table 1.** Statistical analysis.

| Metrics | SOFM Algorithm | LSTM Algorithm |
| --- | --- | --- |
| Number of hops | 28 | 50 |
| Connectivity | 0.99 | 0.99 |
| Coverage | 0.99 | 1.6 |
| Lifetime | 126 | 46.69 |
| ANOVA 1 | $6.04755 \times 10^{-89}$ | $1.11 \times 10^{-16}$ |

The training of the LSTM algorithm is essential for determining the optimal coverage sequence. The fluctuation of the loss in the training process is due to the data sample size as well as the update in the learning weight. During this process, multiple values are explored and then processed to reach the required weight of stabilisation. As illustrated in Figure 5 below, the root mean square error (RMSE) shows a clear trend over iterations. It reached a saturation point after the 40th iteration, suggesting that the network weights had stabilised. This stabilisation is reflected in the improvement of the loss function, which in turn enhances the accuracy of the training.

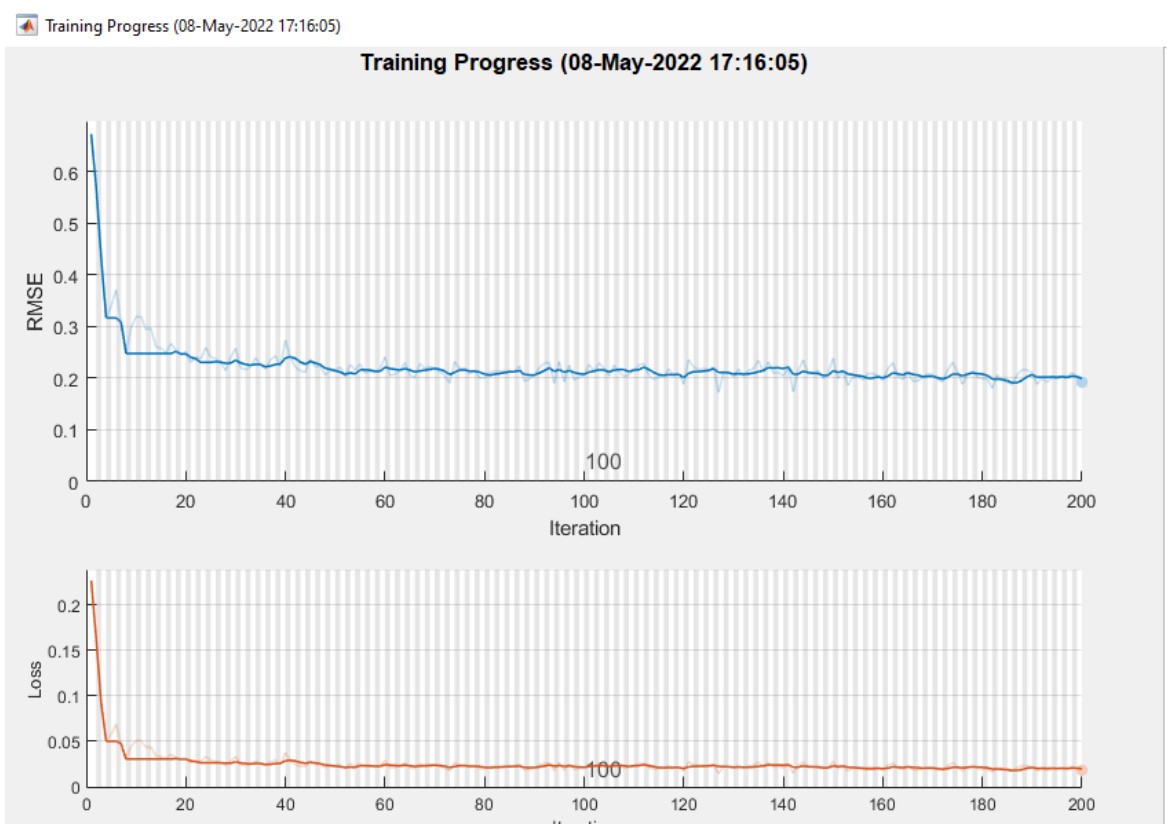

**Figure 5.** Training progress.

### 4.1. All Used Energy

This metric quantifies the total energy consumption of the entire network over the duration of the simulation, continuing until all nodes have completely depleted their energy reserves. The *X* axis corresponds to time measured in round units, while the *Y* axis represents energy consumption, as shown in Figure 6.

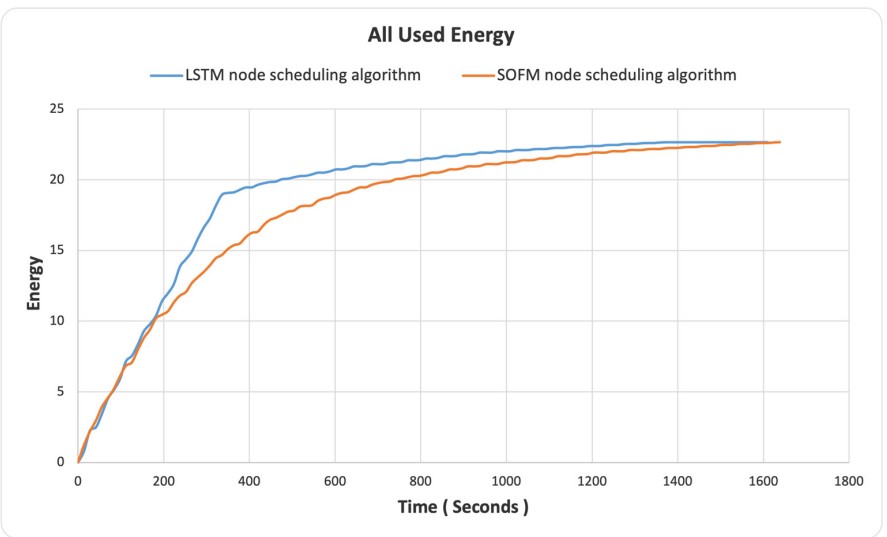

**Figure 6.** All used energy.

While both algorithms initially exhibited similar energy consumption rates at the onset of the simulation, a noticeable distinction emerged as the simulation progressed. Up to the 200th round, the SOFM algorithm consistently consumed less energy overall compared with the LSTM node scheduling algorithm. However, as the simulation advanced, the disparity in energy consumption gradually diminished, approaching near equivalence after the 600th round. This unexpected observation can be attributed to the trade-off design inherent in the LSTM algorithm, where the activation of a maximum number of ON nodes in the WSN plays a role.

It is worth noting that the dependability of a WSN, encompassing aspects like service availability and network reliability, is inherently more time-dependent. This temporal characteristic arises because the WSN's energy depletes over time, in contrast to the spatial changes in terms of geometry. This temporal aspect is exemplified by the energy consumption trend depicted by the blue curve in the LSTM algorithm's performance.

### 4.2. Lifetime of Sensor Nodes

This metric depicts the count of operational sensor nodes within the network throughout the simulation, reflecting the network's longevity. This is visualised on the *Y* axis in Figure 7.

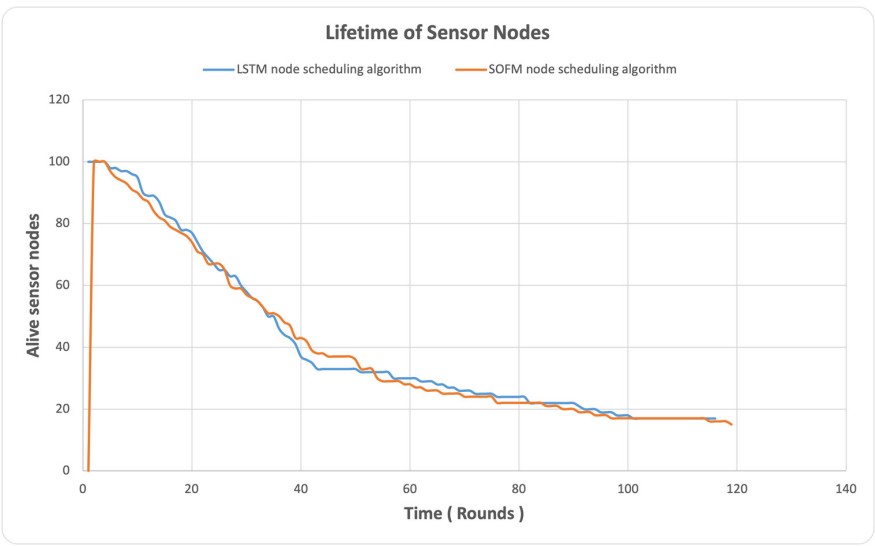

**Figure 7.** Lifetime of sensor nodes.

The plots presented here confirm the findings discussed in Section 4.1, where both algorithms sustained the WSN for a similar number of simulation rounds. As we require a minimum of two input values at times $t − 1$ and t to predict the value at $t + 1$, it is important to note that the LSTM curve, represented by the blue curve in Figure 6, exhibits an initial gap. The Figure 6 clearly shows that the LSTM curve consistently underperformed compared with the SOFM curve, specifically between the 40th and 55th simulation rounds. This discrepancy is attributed to the prevalence of short sequences of nodes that significantly impact the WSN's overall performance in terms of longevity, coverage, and connectivity.

These results align with our hypothesis of identifying a more effective scheduling algorithm for maintaining service availability while minimising the impact on other critical metrics. In this case, we observe that the LSTM algorithm sustained fewer "Alive Sensor Nodes" compared with the SOFM algorithm, although the difference was not overly pronounced throughout the simulation period.

At the 117th round, there were no substantial improvements in the lifetime of the sensor network with the LSTM algorithm. This can be attributed to the training network not providing sufficient energy reserves to carry the network through the subsequent rounds. This limitation is related to the threshold, defined as the minimum energy level at which nodes cease operation, as set in the sensor nodes within the WSN.

### 4.3. Efficiency

The network's bandwidth is determined by the quantity of packets generated within it. Lower bandwidth equates to reduced congestion and fewer data, while higher bandwidth leads to increased congestion and more data. The *X* axis represents the number of rounds, while the *Y* axis represents the number of packets in the network. The primary focus lies on the volume of packets transmitted and received within the WSN.

For the sake of efficiency, the LSTM curve, represented by the blue line in Figure 8, represents information accurately conveyed and received within the LSTM network; this was evident from rounds 10 to 105. The LSTM algorithm consistently demonstrated its superiority over the SOFM algorithm, delivering enhanced efficiency throughout the simulation rounds, until it reached parity towards the end of the simulation. This convergence in efficiency performance is a common observation, as the number of active nodes declines and aligns, regardless of the volume of generated data packets, and their constrained availability ensures identical efficiency. Notably, after the 105th round, the LSTM curve experienced a cessation, primarily attributed to the energy threshold—the point at which nodes cease operation—dictated by the minimum energy configuration within the sensor nodes of the WSN.

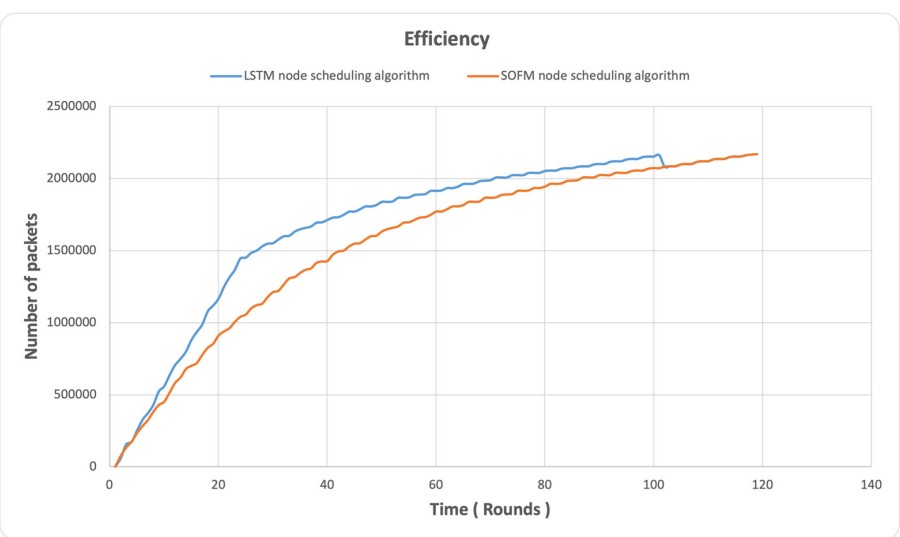

**Figure 8.** Efficiency.

### 4.4. Connectivity

Connectivity quantifies the extent to which nodes are linked within a network. In Figure 9, the *Y* axis represents connectivity performance, gauged by the count of connected nodes. In the simulation, the network comprises 100 nodes. When the connectivity factor reaches 0.98, it signifies that 98 out of the 100 nodes are connected. The *X* axis corresponds to the number of rounds in the simulation.

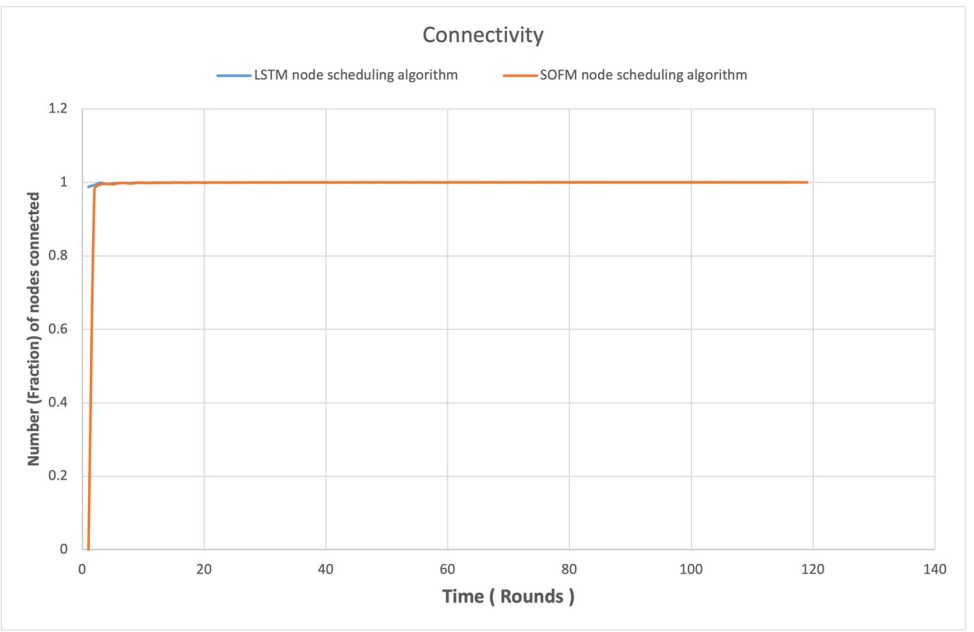

**Figure 9.** Connectivity.

Based on the data presented in Figure 8, the findings support the hypothesis that the newly proposed LSTM algorithm does not significantly surpass nor underperform the SOFM node scheduling algorithm in terms of network connectivity. The primary focus of LSTM is therefore on enhancing service availability while maintaining the other critical metrics, such as efficiency, coverage, and the lifespan of sensor nodes.

In the case of LSTM, several metrics, including the sensor node lifespan, efficiency, and coverage, either exceeded or remained on par with the performance observed with SOFM. This can be attributed to LSTM's efficient temporal energy management and the optimisation of the WSN's topology. However, it is worth noting that when it comes to connectivity, we did not observe substantial differences between the two algorithms, which aligns with our initial hypothesis.

It is important to highlight that the blue curves in Figure 9 representing sensor networks are not directly controlled in terms of message transmission and adjustments by the base station. Instead, they are indirectly managed, which can make these networks susceptible to disturbances.

### 4.5. Coverage

Coverage represents the extent of the tested area in a specific field. In Figure 10, the *Y* axis displays coverage performance, quantified as a numeric value (the fraction of nodes connected in the network). In the simulation, coverage corresponded to the proportion of the total area covered by the network. For instance, with a test field area of 100 m$^2$, a coverage value of 0.8 implies that 80 m$^2$ has been successfully covered. The *X* axis denotes the number of simulation rounds.

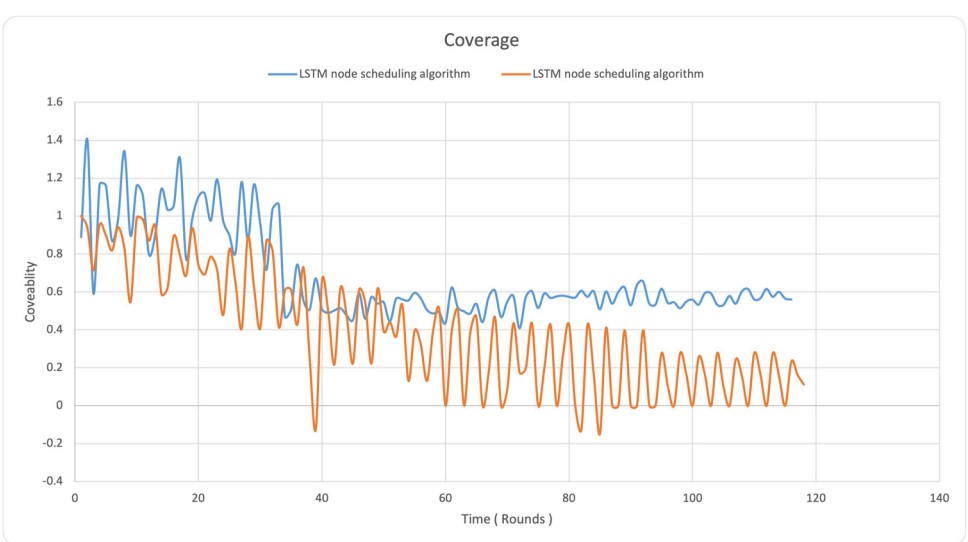

**Figure 10.** Coverage.

The primary objective in this context is to ensure complete coverage of all ON nodes, which means that the sequence of efficient nodes maintaining stable coverage (with minimal fluctuations in the $Y$ axis) must be at least as large as the area being monitored. Consequently, it is imperative to stabilise the coverage and optimise it while considering the network's acceptable lifetime.

It is evident that the coverage started to stabilise from the 39th round and continued in a less fluctuating state until the 116th round. The hypothesis revolves around identifying an efficient coverage pattern and then replicating it throughout the entire network lifecycle using LSTM. The accompanying Figure 10 demonstrates how the set hypothesis is realised, with LSTM values showing greater stability and higher values than SOFM in the latter half of the simulation.

## 5. Conclusions

In this study, we substantiated our hypothesis by discovering an efficient coverage sequence that enhances the overall coverage without compromising network connectivity and longevity. Our algorithm unveiled several challenges within this scheduling algorithms, notably addressing the issue of erratic coverage behaviour. With the application of LSTM, we achieved a more stable coverage performance, especially in the latter half of the network's lifespan. These findings hold significant value for safety-critical WSNs, an area of paramount interest for our research team.

Furthermore, our work demonstrated that using the LSTM algorithm led to higher coverage compared with the SOFM node scheduling algorithm. Another notable advantage of the LSTM node scheduling algorithm is its ability to automate the design process. Initially, LSTM dedicates its efforts and time to discovering an optimal coverage sequence, as illustrated by the fluctuation in the coverage curve depicted in Figure 10. However, once the algorithm identifies effective parameters, it autonomously ensures stable coverage throughout the network's lifetime.

The efficiency of the LSTM node scheduling algorithm was validated by running a training process which included a series of epochs (iterations), aiming to minimise the root mean square error (RMSE) and loss functions, as depicted in Figure 5. This training process constituted 200 iterations to achieve a stable weight state. As depicted in Figure 5, it became evident that after 100 iterations, the RMSE and loss functions converged to similar values and exhibited consistent patterns of fluctuation throughout the remaining iterations. This convergence shows a stable configuration of the LSTM algorithm's weights. If such stability is not achieved, then increasing the number of iterations to enhance the training performance is required.

In the context of future research considerations, we recommend refining the design of the LSTM algorithm to better handle training with a larger dataset, as the current algorithm encountered difficulties after processing just 100 training samples [9]. This limitation is also applicable to large-scale WSNs containing millions of sensor nodes, where the LSTM's memory constraints impede its performance. While transformers present a logical alternative to LSTM, our system's capacity was limited to 100 to 150 nodes, and to reduce computational costs, we opted for LSTM.

**Author Contributions:** Methodology, I.A.-N. and A.L.; Software, I.A.-N.; Validation, I.A.-N.; Formal analysis, I.A.-N. and A.L.; Investigation, I.A.-N. and R.R.; Resources, I.A.-N.; Data curation, I.A.-N.; Writing—original draft, I.A.-N.; Writing—review & editing, I.A.-N., A.L., R.R. and A.K. All authors have read and agreed to the published version of the manuscript.

**Funding:** This research received no external funding.

**Data Availability Statement:** Data are contained within the article.

**Acknowledgments:** The authors express their gratitude to everyone who took part in the accomplishment of this project. A heartfelt appreciation goes out to the Faculty of Science and Technology at Middlesex University for their unwavering support throughout all phases of the research.

**Conflicts of Interest:** The authors declare no conflict of interest.

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
