# Peer review of "A Novel Scheduling Algorithm for Improved Performance of Multi-Objective Safety-Critical Wireless Sensor Networks Using Long Short-Term Memory"

_electronics, doi:10.3390/electronics12234766_

Round 1
Reviewer 1 Report
Comments and Suggestions for Authors
This work presents a new MOO-addressing strategy to improve their ideal QoS. To investigate and provide an energy-efficient scheduling solution, the LSTM model is suggested. The goal of using LSTM is to harmonize/balance WSN parameters for the optimal system output. Epochs (iterations) confirmed the LSTM algorithm's efficiency to meet our service quality goals. LSTM can identify a sequence of events and replicate it as the system's default configuration. Once the good values are discovered, LSTM automatically guarantees service availability and reliability throughout the network lifespan. In general, the paper should be written in correct English. Several key ideas are poorly written and described. Please have the next version proofread by a colleague whose native language is English. Here are some examples of sentences that should be revised.
Comments and Suggestions for Authors
1. Paper must be proofread. Correct up typos, grammar, and language.
2. authors must demonstrate how their approach outperforms others, such as comparative analysis.
3. I propose listing this work's major contributions at the end of the introduction.
4. What are the methods to minimize attacks in WSN?
5. Show explicitly what is “new” in the method you proposed.
6. What scheduling method to improve energy consumption in WSN?
7. How to reduce energy consumption in wireless sensor network?
8. Why is training loss fluctuating?
9. Why LSTM is efficient in sequence Modelling?
10. How does LSTM prove efficient over RNN?

Author Response
Dear Reviewer,
Thanks for your valuable feedback and comments that have been addressed thoroughly. The following are the replies to your comments:
1. All typos, grammar and language issues have been addressed. The entire paper has been proofread by Native Academic English.
2. We have revised the paper accordingly and compared our work with related algorithms from the literature e.g., the SOFM approach. We have run an evaluation against the SOFM node scheduling algorithm and performed statistical analysis as can be shown in Table 1.
3. The main contribution of the paper can be found within lines 126-129.
4. Comment 4 is outside the scope of this paper. The paper’s focus is addressing the multi-objective optimisation problem in WSN. Hence, finding the best trade-off between these objectives namely connectivity, coverage and network lifetime.
5. We proposed the utilisation of a temporal/ time-based model (LSTM) to find a time series of good values, called data points. This data has the maximum coverage, connectivity with efficient energy. This data point has been captured during the training process of the LSTM, and then replicated on the entire network lifetime for a better and efficient performance.
6. and 7. We have used the node scheduling method (Sleep and awake) to find the best coverage and connectivity linked with the optimise energy level in WSN.
8. The fluctuation of the loss in the training process is due to the data sample size as well as the update in the learning weight. During this process, multiple values are explored and then processed to reach the required level of stabilisation. That was mentioned in lines 410 to 412.
9. Because we are using small data samples for training. In our case 100 to 150 sensor nodes.
10. LSTM has proven its efficiency over the RNN because it has higher memory power that better retains long-term dependencies in the data which is perfectly suitable for the MOO problem.
Reviewer 2 Report
Comments and Suggestions for Authors
The manuscript is well structured and the topic is very interesting. Both LSTM algorithms and MOO are now adopted in a large variety of disciplines, from engineering design to software development to economics.
Section 1 offers a good and instructive insight on LSTM, even adding some simple examples, while section 2 provides with a good overview of the recent literature. The research problem and study purpose are further described in section 3.
I do not have technical remarks, as the proposed model seems to perform well on the suggested metrics, but I have several editorial questions and suggestions.
1. Although the paper is overall understandable, a full revision of the English language is needed. Typos, grammatical errors, truncated periods and awkward wording are abundant.
2. The bibliography is puzzlingly minimal.
3. L.198: “ See the coverage see figure 9 we used LSTM to capture the best coverage patterns.” What is this line about? Figure 9 already?
4. Section 3, which starts getting more technical compared to previous introductory sections, is plagued by too many unreadable paragraphs without punctuation.
5. Figure 2: “Memeroy”, “Forgt”.
6. Figure 3 is too small, I recommend increasing the size a bit. Also the font size should be larger, it is not easy to read all the numbers at the moment.
7. Line 328: “The above figure 7.4”.
8. Figure 5: more typos.
9. All the plots in Figures 5 onwards (MS Excel plots?) have low quality, the resolution must be increased.
10. L. 421: “In Figure XXX,”.
11. There is no critical discussion on the limitation of the study; this should appear in a dedicated Discussion section. As in this case this is integrated with Results, perhaps it should be added at the end of Section 4.
12. Also the Conclusions should end with a paragraph that, besides listing future perspectives (as it is already done here), should summarise the main shortcomings and weaknesses of the study.
Comments on the Quality of English LanguageThe English language is quite bad, as observed in my report.
Author Response
Dear Reviewer,
Thanks for your valuable feedback and comments that have been addressed thoroughly. The following are the replies to your comments:
- All typos, grammar and language issues have been addressed. The entire paper has been proofread by Native Academic English.
- We have improved our list of references; however, it is worth mentioning that the application of the LSTM in our problem domain is new and hence there isn’t much published work in the area.
- The paper has been revised and the entire line has been rewritten
- Again, the entire paper has been revised and rewritten so the paragraphs, punctuation and even unreadable figures all were addressed/ replaced with clearer ones.
- The figure has been replaced with the correct spelling legends.
- That has been addressed and the figure resolution has been improved.
- That has been fixed
- That has been addressed
- All resolution of the paper figures has been improved as requested.
- That has been addressed
- We have discussed the limitation of this study in the future work of this paper where we thought that it would be more suitable to be mentioned in that location. “For future work considerations, we can recommend improving the design of LSTM to improve performance over 1000 training samples as currently, the algorithm struggles after 100 training samples [Cheng et al., 2019]. This is also applied for large-scale WSNs (with millions of sensor nodes) as LSTM halts performance in such networks due to its memory constraints. Transformers are the logical replacement for LSTM but in our case, the capacity was 100 to 150 nodes so to reduce the cost of calculation LSTM has been used. This simulation experiment can be emulated in real life by considering a test bed of 20 absolute sensor nodes which can be replicated for 100 to 150 nodes in software, the simulation results can be tested on this test bed using a customised WSN framework that uses MQTT protocol for data transfer - MQTT is an OASIS standard messaging protocol for the IoT.”
- The conclusion section has been rewritten fully to address your previous comments regarding this section.
Reviewer 3 Report
Comments and Suggestions for Authors
In this paper, a novel scheduling algorithm for improved performance of Multi-Objective Safety-Critical WSN Using LSTM is made. Before being published, I suggest some improvements in the presentation and in the content.
The introduction should be improved.
In yours previous analysis [2], [3], [4], more details can be used.
Figure 2, and others, include a legend of the utilized symbols.
Figure 4 should be improved. Some lines are not visible.
Put Figure 6 instead of Figure 7.6.
The legend of the figures should be improved.
Do not put a title inside the figures when the title is in the legend. Include sufficient information in the legend. When a figure has two pictures, place a) and b) and clarify it in the legend.
Improve the references. Add more references related to the work and included in the related works.
Add a table with all inputs of the numerical model.
Use a super index in m2 presented in the work.
Improve the results discussion and more quantitative results should be included in the conclusion.
Add a Nomenclature in the work. For example, in the final of the work.
Comments on the Quality of English LanguageMinor editing of English language required
Author Response
Dear Reviewer,
Thanks for your valuable feedback and comments that have been addressed thoroughly. The following are the replies to your comments:
The introduction has been rewritten addressing your comments.
Regarding the further explanation linked to our previous work [2], [3], and [4] that has been addressed.
All comments regarding figures have been considered and figures resolution has been improved.
The list of references has been improved and more related work from the literature has been considered.
The super index in m2 has been addressed.
Round 2
Reviewer 2 Report
Comments and Suggestions for Authors
The authors have addressed my criticism and the paper can now be published.
As a side note, I advise not submitting a Word document with track changes because it is indeed quite unreadable and time consuming to "decipher".
Reviewer 3 Report
Comments and Suggestions for Authors
In the actual version, in general, all suggestions given by the reviewer was commented.
Comments on the Quality of English LanguageModerate editing of English language required